# Dynamic Evolution of Humoral and T-Cell Specific Immune Response to COVID-19 mRNA Vaccine in Patients with Multiple Sclerosis Followed until the Booster Dose

**DOI:** 10.3390/ijms24108525

**Published:** 2023-05-10

**Authors:** Serena Ruggieri, Alessandra Aiello, Carla Tortorella, Assunta Navarra, Valentina Vanini, Silvia Meschi, Daniele Lapa, Shalom Haggiag, Luca Prosperini, Gilda Cuzzi, Andrea Salmi, Maria Esmeralda Quartuccio, Anna Maria Gerarda Altera, Anna Rosa Garbuglia, Tommaso Ascoli Bartoli, Simonetta Galgani, Stefania Notari, Chiara Agrati, Vincenzo Puro, Emanuele Nicastri, Claudio Gasperini, Delia Goletti

**Affiliations:** 1Department of Neurosciences, San Camillo Forlanini Hospital, 00152 Rome, Italy; serena.ruggieri@uniroma1.it (S.R.); ctortorella@scamilloforlanini.rm.it (C.T.); shaggiag@scamilloforlanini.rm.it (S.H.); lprosperini@scamilloforlanini.rm.it (L.P.); quartuccioe@libero.it (M.E.Q.); galgasi@tiscali.it (S.G.); cgasperini@scamilloforlanini.rm.it (C.G.); 2Department of Human Neurosciences, Sapienza University of Rome, 00185 Rome, Italy; 3Translational Research Unit, National Institute for Infectious Diseases Lazzaro Spallanzani-IRCCS, 00149 Rome, Italy; alessandra.aiello@inmi.it (A.A.); valentina.vanini@inmi.it (V.V.); gilda.cuzzi@inmi.it (G.C.); andrea.salmi@inmi.it (A.S.); annamaria.altera@inmi.it (A.M.G.A.); 4Clinical Epidemiology Unit, National Institute for Infectious Disease Lazzaro Spallanzani-IRCCS, 00149 Rome, Italy; assunta.navarra@inmi.it; 5UOS Professioni Sanitarie Tecniche, National Institute for Infectious Diseases Lazzaro Spallanzani-IRCCS, 00149 Rome, Italy; 6Laboratory of Virology, National Institute for Infectious Diseases Lazzaro Spallanzani-IRCCS, 00149 Rome, Italy; silvia.meschi@inmi.it (S.M.); daniele.lapa@inmi.it (D.L.); annarosa.garbuglia@inmi.it (A.R.G.); 7Clinical Division of Infectious Diseases, National Institute for Infectious Diseases Lazzaro Spallanzani-IRCCS, 00149 Rome, Italy; tommaso.ascoli@inmi.it (T.A.B.); emanuele.nicastri@inmi.it (E.N.); 8Cellular Immunology Laboratory, National Institute for Infectious Diseases Lazzaro Spallanzani-IRCCS, 00149 Rome, Italy; stefania.notari@inmi.it (S.N.); chiara.agrati@opbg.net (C.A.); 9Department of Pediatric Hematology and Oncology, IRCCS Bambino Gesù Children’s Hospital, 00146 Rome, Italy; 10UOC Emerging Infections and Centro di Riferimento AIDS (CRAIDS), National Institute for Infectious Diseases Lazzaro Spallanzani-IRCCS, 00149 Rome, Italy; vincenzo.puro@inmi.it

**Keywords:** SARS-CoV-2, COVID-19, mRNA vaccines, multiple sclerosis, antibody response, T-cell response

## Abstract

This study characterizes antibody and T-cell immune responses over time until the booster dose of COronaVIrus Disease 2019 (COVID-19) vaccines in patients with multiple sclerosis (PwMS) undergoing different disease-modifying treatments (DMTs). We prospectively enrolled 134 PwMS and 99 health care workers (HCWs) having completed the two-dose schedule of a COVID-19 mRNA vaccine within the last 2–4 weeks (T0) and followed them 24 weeks after the first dose (T1) and 4–6 weeks after the booster (T2). PwMS presented a significant reduction in the seroconversion rate and anti-receptor-binding domain (RBD)-Immunoglobulin (IgG) titers from T0 to T1 (*p* < 0.0001) and a significant increase from T1 to T2 (*p* < 0.0001). The booster dose in PwMS showed a good improvement in the serologic response, even greater than HCWs, as it promoted a significant five-fold increase of anti-RBD-IgG titers compared with T0 (*p* < 0.0001). Similarly, the T-cell response showed a significant 1.5- and 3.8-fold increase in PwMS at T2 compared with T0 (*p* = 0.013) and T1 (*p* < 0.0001), respectively, without significant modulation in the number of responders. Regardless of the time elapsed since vaccination, most ocrelizumab- (77.3%) and fingolimod-treated patients (93.3%) showed only a T-cell-specific or humoral-specific response, respectively. The booster dose reinforces humoral- and cell-mediated-specific immune responses and highlights specific DMT-induced immune frailties, suggesting the need for specifically tailored strategies for immune-compromised patients to provide primary prophylaxis, early SARS-CoV-2 detection and the timely management of COVID-19 antiviral treatments.

## 1. Introduction

Multiple Sclerosis (MS) is a neurodegenerative and autoimmune disease of the central nervous system, counting as the first non-traumatic cause of disability in young adults in Western countries [1]. During the last two decades, the accrual of the neurological deficit in patients with MS (PwMS) has been dramatically halted by the introduction of Disease Modifying Treatments (DMTs) that mainly target the individual immune response, potentially impairing the humoral- and cell-mediated responses to infections and vaccinations [2]. In this context, during the COronaVIrus Disease 2019 (COVID-19) pandemic, great concern has been expressed toward the management of PwMS. Firstly, most issues regarded the increased risk of worse outcomes of Severe Acute Respiratory Syndrome Coronavirus 2 (SARS-CoV-2) infection in PwMS [3] while, later on, the likely compromised response to vaccination against it [4].

Currently, more than 20 vaccines based on different platforms have been approved globally, and 11 of them have been listed and authorized by the WHO for emergency use [5]. Global vaccination has led to a reduction in transmission, disease severity, hospitalizations and deaths [6,7].

The available vaccine platforms are based on messenger RNA (mRNA) (BNT162b2, mRNA-1273), adenoviral vectors (ChAdOx1-nCoV-19, Covishield, Ad26CoV2.S, Ad5-nCoV), inactivated virus (Corona Vac, BBIBP-CorV, BBV152) and recombinant proteins (Nvx-CoV-2373, Covovax) [8]. In Italy, 84.7% of the population completed the vaccine cycle and the majority of the individuals (90.9%) have been vaccinated with the mRNA-based platform [9]; for this reason, this study has been focused on this kind of vaccine. 

The available vaccines are designed to induce specific immunity against the spike protein [10], which is critical for SARS-CoV-2 binding and cell entry. Immunoglobulin (Ig)G and IgM anti-receptor-binding domain (RBD) antibodies, neutralizing antibodies and CD4^+^ and CD8^+^ T-cell responses [11,12,13] are elicited by these intramuscular-based vaccines differently from the mucosal-based vaccines that are also able to induce IgA [8]. Besides the ability to neutralize the virus, antibodies are important for several functions, and T cells can control viral replication by killing viral-infected cells and modulating the B-cell responses and, consequently, antibody production.

Although the vaccination reduced the epidemic worldwide, breakthrough cases have been frequently reported due to the raising of SARS-CoV-2 variants [14,15,16], concomitant easing of distancing measures and waning of vaccine immunity over time [17,18,19,20,21,22,23]. This led the regulatory agency to suggest a third dose of an mRNA vaccine, starting with the most immunologically fragile subjects [24], based on evidence that titers of antibodies can decline over time but can be boosted by a third vaccine dose [25,26].

We recently showed that mRNA vaccines induced both humoral- and cell-mediated specific immune responses in the majority of PwMS after the completion of a two-dose vaccination; however, a different grade of response was demonstrated according to the specific DMT [27,28,29,30,31]. Based on new evidence and recommendations, it would be crucial in this fragile population to understand how patients’ immune response to the vaccine can vary over time, and whether it is efficiently boosted by a third dose.

Several reports have tried to address these points, often focusing on PwMS treated with specific compounds [32,33,34,35] or only evaluating the humoral- or T-cell response [36,37,38,39,40,41]. The aim of this present study was to evaluate both the humoral- and T-cell immune responses over time and, following the third dose of COVID-19 vaccination in PwMS undergoing different treatments, comparing them with the response obtained in health care workers (HCWs). 

## 2. Results

### 2.1. Characteristics of the Study Population

Out of 144 PwMS screened for this study, 10 subjects were excluded from analyses: seven had signs of previous SARS-CoV-2 infection verified by anti-nucleoprotein-immunoglobulin G (anti-N-IgG) levels, two were not treated at the time of enrolment and one was treated with a drug different from all the other patients included. From the established cohort of 134 patients at baseline (T0, 2–4 weeks after the second dose), 13 individuals were lost at follow-up, finally resulting in a total of 121 patients (Figure 1A,B). A total of 99 HCWs were prospectively enrolled as the healthy control group. 

Among the enrolled subjects, 105 PwMS and 89 HCWs were assessed after 6 months (T1) from the first mRNA vaccine dose, while 82 PwMS and 38 HCWs were evaluated 4–6 weeks after the booster dose (T2). To study the overtime evolution of the immune responses, 64 PwMS and 25 HCWs providing blood samples at all three timepoints were analyzed (Figure 1B). 

The demographic and clinical data are summarized in Table 1. The two cohorts significantly differed by age (*p* = 0.001) and showed a female predominance of about 70% due to the three times higher frequency of MS in women than in men [42] and the high prevalence of females among HCWs. Within the PwMS cohort, 34 subjects were treated with ocrelizumab, 35 with fingolimod, 20 with cladribine and 32 with interferon (IFN)-β. The lymphocyte count was significantly decreased in patients treated with fingolimod compared to those treated with other DMTs (*p* < 0.0001).

### 2.2. Antibody and Spike-Specific T-Cell Response in PwMS and HCWs at T1

At T1, all the HCWs maintained a detectable anti-RBD-IgG response, whereas only 65.7% of PwMS had a positive response. Although most PwMS showed an antibody response, the seroconversion rate and the quantitative-specific responses significantly varied among DMTs (*p* < 0.0001) (Table 2 and Figure 2A).

Regarding the qualitative response, ocrelizumab- and fingolimod-treated patients showed the lowest seroconversion rates, 20.7% and 58.1%, respectively, compared with HCWs (*p* < 0.0001 for both) (Table 2). In contrast, all the patients undergoing cladribine or IFN-β seroconverted. Regarding the quantitative response, lower antibody titers were found in PwMS treated with ocrelizumab (*p* < 0.0001), fingolimod (*p* < 0.0001) and cladribine (*p* = 0.002) compared with HCWs. No significant difference was found for IFN-β-treated subjects (Figure 2A). 

Despite the majority of fingolimod-treated patients showing an anti-RBD-IgG response, the neutralizing activity was only detected in 4/31 (12.9%) individuals and at lower titer than HCWs (*p* < 0.0001) (Figure 2B). Conversely, 20/27 (74.1%) IFN-β-treated patients presented neutralizing antibodies; however, their titers were significantly lower than those of HCWs (*p* < 0.0001). Significant correlations were found between neutralizing antibodies and anti-RBD-IgG titers in both HCWs (rho = 0.653, *p* < 0.0001) and PwMS treated with fingolimod (rho = 0.511, *p* = 0.003) or IFN-β (rho = 0.403, *p* = 0.037).

To account for the demographic and clinical factors potentially affecting the immune response, univariable regression modeling was performed (Appendix A) and the covariates with *p* < 0.1 were entered into the multivariable model (Table 3). 

Age and sex did not appear to influence the anti-RBD-IgG response but, after adjusting for the weeks elapsed from the second vaccine dose, the differences observed among DMTs and HCWs only persisted for the patients treated with fingolimod or ocrelizumab (Table 3). Moreover, after adjusting for the variables included in the final model, with respect to the patients treated with IFN-β, those treated with cladribine showed a similar average anti-RBD-IgG level (IRR: 0.78, 95%CI: 0.29 to 2.11, *p* = 0.624), while the patients treated with fingolimod or ocrelizumab had a significantly lower level of 77% (IRR: 0.23, 95%CI: 0.10 to 0.53) (*p* = 0.001) and 96% (IRR: 0.04, 95%CI: 0.02 to 0.12) (*p* < 0.001) (Table 3). 

Regarding the T-cell response, at T1, 60% of PwMS showed an IFN-γ-specific T-cell response compared with 98.9% of HCWs (Table 2). Within the MS cohort, significantly different proportions of T-cell responders were found among DMTs (*p* < 0.0001). In particular, the response rates of fingolimod- (3.2%), cladribine- (61.1%) and IFN-β-treated patients (85.2%) were significantly reduced compared with HCWs (*p* < 0.0001, *p* < 0.0001, *p* = 0.002, respectively) (Table 2). Differently, patients undergoing ocrelizumab (28/29, 96.5%) showed a T-cell response rate similar to the controls (88/89, 98.9%). 

Regarding the quantitative response, all PwMS had significantly reduced IFN-γ-specific levels compared with HCWs, regardless of the ongoing treatment (Figure 2C), but, after adjusting for weeks and age, PwMS treated with IFN-β did not show statistically significant differences (Table 3). Compared with the IFN-β-treated patients, those under fingolimod, after adjusting for the lymphocytes count, were estimated to have an average 98% lower spike-specific IFN-γ level (IRR: 0.02, 95%CI: 0.01 to 0.04) (*p* < 0.001); differently, cladribine- and ocrelizumab-treated patients had, respectively, IFN-γ levels that were 32% lower than the IFN-β-treated patients (IRR: 0.68, 95%CI: 0.26 to 1.75) (*p* = 0.421) and 13% higher (IRR: 1.13, 95%CI: 0.49 to 2.57) (*p* = 0.778), without statistical significance.

### 2.3. Antibody and Spike-Specific T-Cell Response in PwMS and HCWs at T2

The differences observed between the two cohorts at T1 were mostly preserved or lost in some cases at T2. Indeed, 78% of PwMS showed an antibody response compared with 100% of HCWs. Only patients treated with ocrelizumab maintained a proportion of responders (5/22, 22.7%), significantly lower than the HCWs (*p* < 0.0001), whereas all cladribine- and IFN-β-treated patients, and almost all fingolimod-treated patients (29/30, 96.7%), seroconverted (Table 4). 

Regarding the quantitative response, lower anti-RBD-IgG titers persisted in ocrelizumab- and fingolimod-treated patients compared with HCWs (*p* < 0.0001 for both) (Figure 2D). In contrast, in patients under cladribine or IFN-β, the booster dose restored anti-RBD-IgG titers to a similar level to HCWs (Figure 2D). These differences compared to HCWs were confirmed in the regression analysis and estimated for the patients under IFN-β even higher levels than HCWs (Table 3).

Compared with the IFN-β-treated patients, after adjusting for the variables included in the multivariable model, the cladribine-treated patients were estimated to have an RBD-IgG level that was on average 50% lower (IRR: 0.50, 95%CI: 0.08 to 3.20), albeit not statistically significant (*p* = 0.465), whereas similar estimations were obtained for those under fingolimod or ocrelizumab with, respectively, 96% (IRR: 0.04, 95%CI: 0.01 to 0.16) and 97% (IRR: 0.03, 95%CI: 0.01 to 0.22) lower anti-RBD-IgG levels (for both *p* < 0.001). 

The anti-RBD-IgG titers significantly correlated with the neutralizing antibodies in both the HCWs (rho = 0.491, *p* = 0.008) and PwMS treated with fingolimod (rho = 0.832, *p* < 0.0001) or IFN-β (rho = 0.806, *p* < 0.0001). Consistent with the increasing seroconversion rate, neutralizing antibodies were detected in a higher percentage of fingolimod-treated patients (14/30, 46.7%), albeit maintaining lower titers than those of HCWs (*p* < 0.0001) (Figure 2E). Differently, the IFN-β-treated patients achieved neutralizing titers equal to or similar to the controls after the booster dose.

Additionally, the differences between the two cohorts observed for the T-cell response were partially preserved at T2. Indeed, 59.7% of PwMS mounted an IFN-γ-specific T-cell response compared with 100% of HCWs (Table 4). Both the fingolimod- (1/30, 3.3%) and cladribine-treated patients (8/10, 80%) maintained a proportion of responders significantly lower than the controls (*p* < 0.0001 and *p* = 0.005, respectively) (Table 4). Regarding the quantitative response, the ocrelizumab- and IFN-β-treated patients achieved IFN-γ levels comparable to those of HCWs, whereas significantly lower levels persisted in the fingolimod- and cladribine-treated patients (*p* < 0.0001 for both) (Figure 2F). After the application of the regression model, all the patients, regardless of DMTs, showed reduced IFN-γ levels compared with the HCWs, although less marked for the IFN-β-treated patients (Table 3). Within the PwMS cohort, the patients treated with cladribine or fingolimod, after adjusting for the lymphocytes, were estimated to have, respectively, an average of 77% (IRR: 0.23, 95%CI: 0.08 to 0.72) (*p* = 0.012) and 99% (IRR: 0.01, 95%CI: 0.001 to 0.01) (*p* < 0.001) significantly lower spike-specific IFN-γ levels than the IFN-β-treated patients, while those under ocrelizumab showed levels similar to the IFN-β-treated patients (IRR: 0.90, 95%CI: 0.38 to 2.15) (*p* = 0.814). 

Overall, 32/82 (39%) of PwMS were full responders as they mounted both an antibody and cell-mediated response after the booster dose. Only one subject was a non-responder, whereas 59.8% (49/82) of the subjects were partial responders (having at least one of the responses). In particular, 39% (32/82) of the subjects mounted only the humoral response, whereas 20.8% (17/82) only mounted a T-cell response (Figure 2G). Stratifying the results according to the therapy, we found that the majority of cladribine- (80%) and IFN-β-treated patients (90%) were full responders. As expected, most of the ocrelizumab- (77.3%) and fingolimod-treated patients (93.3%) were partial responders, showing only a T-cell or humoral response, respectively. 

### 2.4. Temporal Evolution of the Antibody Response to COVID-19 Vaccination in Both HCWs and PwMS 

The antibody response to the SARS-CoV-2 vaccination was evaluated in the PwMS (*n* = 64) and HCWs (*n* = 25) longitudinally sampled from T0 to T2. In the HCWs, the anti-RBD-IgG titers significantly decreased from T0 to T1 (*p* = 0.0001, a 20-fold decrease), whereas a significant 26-fold increase was found from T1 to T2 (*p* < 0.0001) (Figure 3A). Nevertheless, no variation was observed in the proportion of antibody responders; all the HCWs showed a positive antibody response regardless of the timepoint considered (Table 5).

Similarly, in the MS cohort, the anti-RBD-IgG titers significantly decreased from T0 to T1 (*p* < 0.0001, a 4-fold decrease), and a significant 20-fold increase in the median titer was observed from T1 to T2 (*p* < 0.0001) (Figure 3B). Interestingly, in PwMS, a significant negative correlation was found between the percentage of decay from T0 to T1 and the percentage of increase from T1 to T2 (rho = −0.574, *p* < 0.0001), meaning that PwMS with a higher decrease also presented a greater antibody increase after the booster dose. A similar trend was also observed in HCWs, albeit not significant (rho = −0.389, *p* = 0.054) probably due to the small sample size analyzed. Consistent with the waning of the median titer, the seroconversion rate significantly differed over time in PwMS (*p* < 0.0001); a lower antibody rate was observed at T1 (37/64, 57.8%) compared to T0 (49/64, 76.6%) and T2 (48/64, 75%) (Table 5). 

In PwMS, the booster dose showed a boosting effect on the serological response as it promoted a significant 5-fold increase of the anti-RBD-IgG titers compared with T0 (*p* < 0.0001) (Figure 3B). However, the number of seroconverted patients was comparable to that observed after only the first vaccination cycle (T0: 76.6% vs. T2: 75%).

Stratifying the MS cohort according to DMTs, we found that the fingolimod and IFN-β-treated patients showed significant variations, in terms of median titer, for the antibody response (Figure 3C). Regarding the proportion of the responders, all the cladribine and IFN-β-treated patients seroconverted independently of the timepoint considered, whereas the patients treated with fingolimod showed a lower seroconversion rate at T1 (15/26, 57.7%) compared to both T0 (23/26, 88.5%) and T2 (25/26, 96.2%) (Figure 3C). As expected, the majority of the ocrelizumab-treated patients failed to induce an antibody response. 

### 2.5. SARS-CoV-2-Spike-Specific T-Cell Response Persists over Time in Both Cohorts

Concomitantly with the antibody response, we monitored the T-cell response to the SARS-CoV-2 vaccination. In HCWs, the booster dose elicited a higher IFN-γ-spike-specific response (2.7-fold) compared to T1 (*p* = 0.0089) (Figure 3D). Moreover, all the HCWs showed a positive T-cell response that persisted regardless of the time elapsed since vaccination (Table 5). Similarly, in PwMS, the IFN-γ levels differed over time showing a significant 3.8-fold increase from T1 to T2 (*p* < 0.0001) (Figure 3E and Table 5). In both HCWs and PwMS, the T-cell response was also boosted compared to T0; although, this difference resulted in statistical significance only in PwMS (*p* = 0.013), probably due to the small sample size of HCWs analyzed (Figure 3E). Despite the increasing IFN-γ levels, we did not observe any significant modulation in the number of responders (Table 5). 

Stratifying PwMS according to therapy, we found that ocrelizumab-treated patients showed significantly higher IFN-γ levels at T2 compared to T0 (*p* = 0.001) and T1 (*p* = 0.0002), whereas no significant modulations were reported for cladribine- or IFN-β-treated patients (Figure 3F). All the fingolimod-treated patients failed to induce a T-cell response even after the booster dose. The only subject under fingolimod who scored positive at T0 recovered the response after the booster dose. 

## 3. Discussion

This study provides detailed evidence of the decreased immune response 24 weeks after the completion of the first SARS-CoV-2 mRNA vaccine course and a boosting effect of both humoral- and cell-mediated responses after an additional vaccine dose in a prospective cohort of PwMS compared with HCWs. 

Most current studies on this topic have only considered the humoral response elicited by the vaccine rather than evaluating the cellular one [36,37,41,43] or their interplay. Indeed, it is well established how a combined humoral and cell response is fundamental to halt the severity of COVID-19 [12,44] and it may be iatrogenic impaired in those subjects affected by an immune-mediated inflammatory disease undertaking immunotherapy [45]. To date, most works have specifically focused on anti-CD20 treatment rather than including other DMTs [19,34,35,46]. 

Moreover, the results may vary according to the methodology and the consistency of the approach used throughout the longitudinal samples, and the time elapsed between vaccination and immunological assessment [38,47]. By closely following the same population of PwMS and HCWs, and using consistent methodology at each timepoint, we demonstrated that the waning of the humoral response is more pronounced, both in terms of magnitude and response rate, in PwMS undergoing ocrelizumab and fingolimod than in those treated with cladribine and IFN-β compared with HCWs. A markedly lower humoral response in patients undergoing fingolimod and anti-CD20 therapies was also recently reported by Sainz de la Maza et al. [31]. This finding is coherent not only with the level of response detected soon after the first vaccine course [27] but also with the mechanism of action of both treatments. Ocrelizumab impacts the IgM and IgG levels throughout the treatment duration and this decrease is related to a higher risk of infections with worsened outcomes [4,48]. Conversely, the effect of fingolimod on antibody production can be mediated, as demonstrated in the animal model, in a T-cell-dependent manner [49]. Despite the main mechanism of action, both drugs seem to affect the humoral and cellular responses [50,51], eventually converging on the effect on CD19^+^ lymphocytes in predicting the response to anti-SARS-CoV-2 vaccination, as it has been observed in patients with hematological malignancies [52]. This is further confirmed by our results showing how the differences observed for the T-cell response between the two cohorts, and according to DMTs, persist over time (T1 vs. T0) and are still more prominent in fingolimod-treated patients.

Our findings show that a booster dose promptly increases the humoral response in PwMS even though lower titers persist compared with HCWs. However, it is relevant to note that the fold increase in the anti-RBD-IgG response was greater in PwMS than in HCWs. This led to the restoration of the humoral response that decreased over time in the cohort, except for most ocrelizumab-treated PwMS who did not show seroconversion at any timepoint. This is consistent with several observations reported to date not only in MS [32,34,37,43] but also across several conditions in which anti-CD20 treatments are prescribed [45]. Interestingly, we showed that the spike-specific T-cell response persists over time and it was even reinforced in ocrelizumab-treated subjects after a third vaccine dose, confirming previous results [53,54]. This acquires the outmost meaning, considering recent evidence showing that vaccine-induced T-cell responses are only poorly influenced by the mutations associated with the Omicron variant in the general population [55] and also in ocrelizumab-treated PwMS [33,56]. Therefore, subjects undergoing ocrelizumab can at least implement a cell-mediated response in case of SARS-CoV-2 infections. 

The all reported pieces of evidence combined together suggest that B-cell-depleting therapies may intensify the predictable humoral waning. The type of COVID-19 vaccine can also differently affect the immune response and the risk of severe breakthrough infections. Indeed, PwMS treated with fingolimod who received two doses of inactivated-virus-based vaccines showed a significantly higher antibody production if the boost was done with an mRNA-based vaccine instead of the inactivated-virus-based vaccines [57]. A surveillance study conducted in the United States also showed that among the individuals who were fully vaccinated with different comorbidities, the recipients of mRNA vaccines had a lower risk of severe breakthrough SARS-CoV-2 infection than those vaccinated with a vector-based vaccine [58].

Regarding the clinical outcomes in PwMS, recent extensive work has highlighted how the risk of breakthrough SARS-CoV-2 infections is mainly associated with reduced levels of the virus-specific antibody response and can be attenuated by a third vaccine dose [59]. However, the clinical correlation with the cellular response against symptomatic infection has not been explored. Nevertheless, most infections (69.3%) showed mild to moderate severity that did not need treatment [51]. In this respect, we recently showed that the booster vaccine dose further increases both the CD4^+^ and CD8^+^ effector memory T cells (T_EM_) and the CD8^+^ terminally differentiated memory T cells (T_EMRA_) in PwMS [60]. The increase of memory T cells might prevent the onset of severe COVID-19 disease as they expand if re-challenged and contribute to prompt immune responses limiting the initial viral replication, including that of the current variants [61], and spread in the host. Indeed, there was a significant reduction in the hospitalization rate after full vaccination (11.9% vs. 3.9%) compared to the pre-vaccination time for all DMTs, including fingolimod but not ocrelizumab [51]. Together with our findings, this underlines the need for a mitigation strategy for the PwMS still at risk of hospitalization despite the full vaccine course, referring them to available and still effective, although only partially, treatments [62]. 

### 3.1. Limitations of the Study

Some limitations are acknowledged. Firstly, the small sample size of patients that completed all three timepoints might have impaired the emergence of small but significant differences among the treatment groups. However, at each timepoint, every DMT was fairly represented, particularly those with an already known compromised response (i.e., ocrelizumab and fingolimod) that needed to be better characterized. Secondly, we lack the untreated MS group. We did not report the correlation with the blood lymphocyte count nor with the timing between the last infusion, but it was already explored in our previous work [27] and across the literature [47]. However, binomial regression models were used to account for demographic and potential clinical confounding for the immune response modulations in PwMS.

### 3.2. Conclusions

In conclusion, PwMS on DMTs experience a decline of humoral- and cell-mediated immune responses that can be strengthened by the third booster dose of the SARS-CoV-2 mRNA vaccine. However, despite vaccination, the PwMS treated with ocrelizumab and fingolimod show reduced humoral- and T-cell specific immune responses, respectively, suggesting the need for specifically tailored strategies for immune-compromised patients to provide primary prophylaxis, early SARS-CoV-2 detection and the timely management of COVID-19 antiviral treatments [63,64,65]. 

## 4. Materials and Methods

### 4.1. Study Population

This longitudinal prospective study was conducted on patients with a diagnosis of MS, who were enrolled at the MS Centre of the Department of Neurosciences of San Camillo Forlanini Hospital (Rome, Italy). Inclusion criteria for their enrollment were the following: (1) diagnosis of MS according to 2017 revisions of McDonald criteria [66]; (2) ongoing DMT with ocrelizumab, fingolimod, cladribine or IFN-β for at least 6 months prior to the enrollment; and (3) completion of the first SARS-CoV-2 vaccination schedule with an mRNA vaccine (BNT162b2 or mRNA-1273) for at least 6 months and/or administration of the booster dose within 4–6 weeks prior to the study entry. For ocrelizumab- and cladribine-treated patients, the COVID-19 vaccination was scheduled based on the last DMT administration following the recommendations of both the Italian and European Academy of Neurology. In particular, ocrelizumab was provided with a delay of at least 3 months from vaccination. Neither cladribine, IFN-β nor fingolimod was discontinued for vaccination. As the healthy control group, a convenient sample of HCWs without any medical condition conferring immunosuppression, and who have undergone SARS-CoV-2 vaccination, was enrolled at the National Institute for Infectious Diseases (INMI)-Lazzaro Spallanzani (Rome, Italy). A portion of HCWs was previously also involved as the control group [26,60]. Exclusion criteria for the enrolled individuals were the following: previous SARS-CoV-2 infection, HIV infection and age <18 years. 

Human study protocols were approved by the Ethical Committee of “L. Spallanzani” National Institute of Infectious Diseases (INMI)-IRCCS (approval numbers 297/2021, 247/2021 and 319/2021) and followed the ethics principles for human experimentation in agreement with the Declaration of Helsinki. All participants signed a written informed consent form before the study procedures.

### 4.2. Study Procedures

From an established cohort previously evaluated at baseline (T0, 2–4 weeks after the second dose) [27], participants were recruited after 24 weeks (T1) from the first mRNA vaccine dose and 4–6 weeks after the booster dose (T2) (Figure 1A). Blood sampling and laboratory procedures were performed following a standardized protocol routinely used [67,68]. Blood samples from PwMS were collected in lithium heparinized tubes (BD Vacutainer, Becton Dickinson, Florence, Italy, Cat. 367526) at the MS Center of San Camillo Forlanini Hospital, then transported to INMI, where they were processed within 2 h of collection. The same protocol was performed for HCWs samples. Immune-based assays and lymphocyte counts were performed within 1 week from the time of sample collection.

### 4.3. Anti-SARS-CoV-2 Antibody Testing

The antibody response was assessed by measuring both anti-nucleoprotein-immunoglobulin G (Anti-N-IgG) and anti-receptor-binding domain (RBD)-IgG as per manufacturer’s instructions (Architect^®^ i2000sr Abbott Diagnostics, Chicago, IL, USA). Anti-N-IgG were reported as index value [(sample (S)/Cutoff (CO)] and titers ≥1.4 were considered positive. Anti-RBD-IgG were expressed as binding antibody units (BAU)/mL and indicated as positive if ≥7.1.

Neutralizing antibodies were assessed by the micro-neutralization assay (MNA) previously described [69]. For the assay, the SARS-CoV-2/Human/ITA/PAVIA10734/2020 (isolated in March and provided by Fausto Baldanti, Pavia, Italy) was used. The neutralization titer was reported as the reciprocal of serum dilution (MNA_90_) representing the highest serum dilution able to inhibit at least 90% of the cytopathic effect (CPE). The threshold value was set at 1:10 and the first dilution was tested.

### 4.4. IFN-γ Release Assay (IGRA)

To evaluate the IFN-γ-specific T-cell response, whole blood was stimulated with a peptide mix spanning the entire sequence of SARS-CoV-2 Wuhan spike protein (PepTivator^®^ Prot_S1, Prot_S, and Prot_S+, Miltenyi Biotec, Bergisch Gladbach, Germany, Cat. 130-127-048, Cat. 130-126-701 and Cat. 130-127-312, respectively) and incubated for 16–24 h at 37 °C, as previously reported [67,68]. For the stimulation, a unique pool (spike) including equal amounts of the three peptide pools at a final concentration of 0.1 µg/mL was used. Staphylococcal enterotoxin B (SEB) (Merck Life Science, Milan, Italy, Cat. S4881) was used at 200 ng/mL as the positive control. After incubation, plasma was harvested and stored at −20 °C or −80 °C until further use. IFN-γ levels were quantified using the ELLA Simple Plex Human IFN-gamma (3rd Gen.) Assay (Bio-Techne, Minneapolis, MN, USA, Cat. SPCKB-PS-002574) and reported after subtracting the unstimulated-control value. The detection limit of the assay was 0.17 pg/mL. The IFN-γ threshold value was set at 16 pg/mL to define a positive responder.

### 4.5. Statistical Analysis

Data were analyzed using GraphPad software version 9.3.1 (GraphPad Prism, San Diego, CA, USA). Continuous variables were reported as the median and interquartile range (IQR), whereas categorical variables were expressed as count and proportion. The following non-parametric inference tests were used: Kruskal–Wallis test for comparisons among groups; Mann–Whitney U-test for pairwise comparisons (for unpaired data) and Chi-square or Cochran’s Q test to compare categorical variables (unpaired and paired data, respectively). Correlations were assessed by the non-parametric Spearman’s rank test (rho). Negative binomial models were used: (i) to explore differences in levels of anti-RBD-IgG and T-cell responses among DMTs after administration of the second vaccine dose at T1 and after the booster dose T2, and (ii) to account for demographic and clinical potential confounding for the immune response modulations in PwMS. Univariable regression modeling was performed and covariates with *p* < 0.1 were entered in the final multivariable models. For the analysis, dependent variables (i.e., anti-RBD-IgG and IFN-γ levels) and covariates were included: sex, age, treatments, lymphocyte count, body mass index (BMI), disease and treatment duration, EDSS score and weeks elapsed from the second or booster dose. MS phenotype (relapsing-remitting or primary-progressive) was not included in the models because it showed unacceptably wide range of uncertainty due to the high collinearity, especially with treatments (variance inflation factor = 16.4). Negative binomial models were also used to compare differences among treatments compared to HCWs after evaluation of the potential confounders available: age and sex. Negative binomial regression was performed with the use of Stata (StataCorp. 2021. Stata Statistical Software: Release 17. TX: StataCorp LLC, College Station, TX, USA). A two-tailed *p* value < 0.05 was considered significant except for subgroup analyses where Bonferroni correction was applied (α/4 = 0.0125).

## Figures and Tables

**Figure 1 ijms-24-08525-f001:**
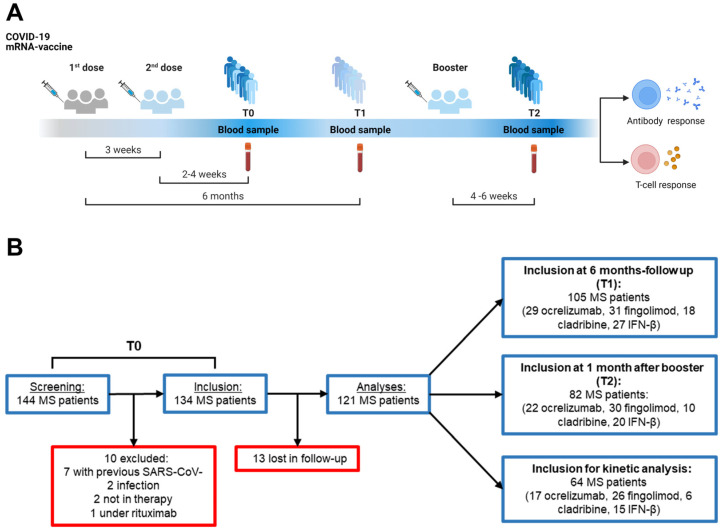
Study design and flow chart of the enrolled subjects. (**A**) Timeline depicting sample collection in relation to the COVID-19 vaccination schedule. (**B**) Participant flow chart showing the patients recruited after 2–4 weeks from the second vaccine dose (T0), after 6 months from the first dose (T1) and 4–6 weeks after the booster (T2). Abbreviations: COVID-19, COronaVIrus Disease 2019; IFN, interferon; MS, multiple sclerosis. Created with BioRender software.

**Figure 2 ijms-24-08525-f002:**
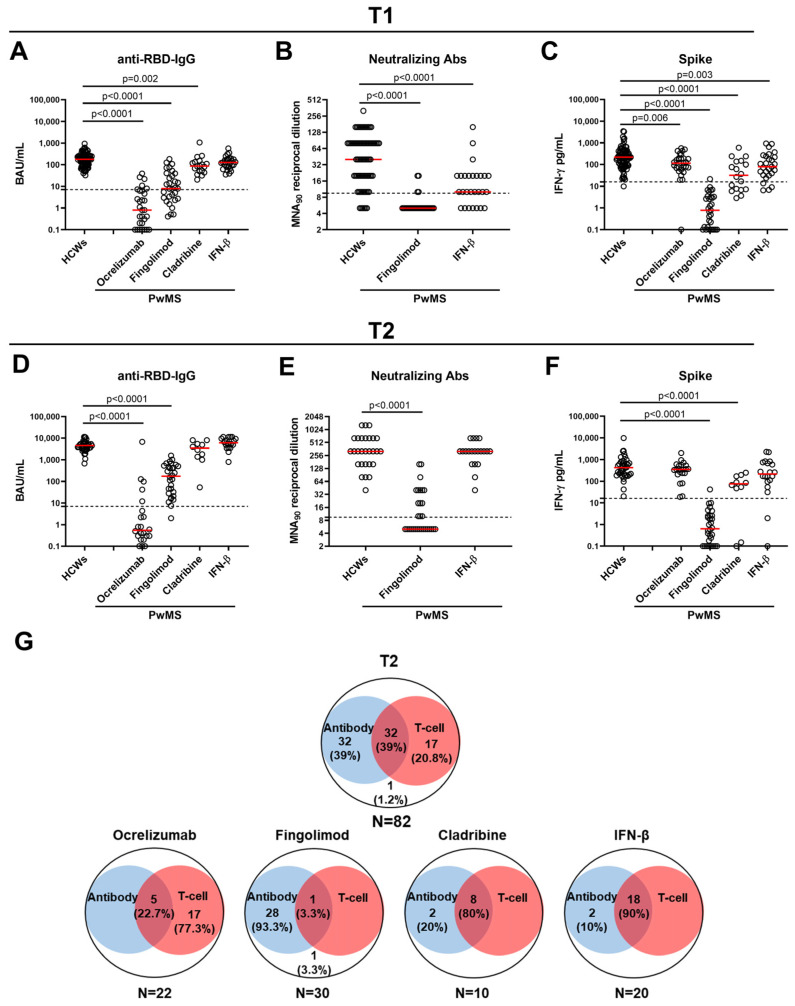
Humoral and IFN-γ T-cell response in HCWs and PwMS before (T1) and after the COVID-19 vaccine booster (T2). (**A**–**C**) Anti-RBD-IgG, neutralizing antibodies and spike-specific IFN-γ T-cell response detected by IGRA were evaluated after 6 months from the first vaccine dose (T1) in 89 HCWs and 105 PwMS. PwMS were stratified based on the undertaken treatment: ocrelizumab (*n* = 29), fingolimod (*n* = 31), cladribine (*n* = 18) and IFN-β (*n* = 27). (**D**–**F**) Anti-RBD-IgG, neutralizing antibodies and spike-specific IFN-γ T-cell response detected by IGRA were evaluated after 4–6 weeks from the booster dose (T2) in 38 HCWs and 82 PwMS. PwMS were stratified based on the undertaken treatment: ocrelizumab (*n* = 22), fingolimod (*n* = 30), cladribine (*n* = 10) and IFN-β (*n* = 20). Anti-RBD-IgG and neutralizing antibodies were measured in serum and expressed as binding antibody units (BAU)/mL and reciprocal of dilution (MNA_90_), respectively. IFN-γ levels were quantified by automatic ELISA in plasma harvested from stimulated blood samples and reported after subtracting the unstimulated control value. Medians were indicated by red horizontal lines. Kruskal–Wallis test was performed to compare groups with Bonferroni correction (α/4) and *p* < 0.0125 was considered significant. Dashed lines identify the cut-off of each test (anti-RBD-IgG: 7.1 BAU/mL; MNA_90_: 10 reciprocal dilution and spike: 16 pg/mL). (**G**) Venn diagrams show the number of responders for antibody and T-cell response. Abbreviations: PwMS, patients with multiple sclerosis; RBD, receptor-binding domain; IgG, immunoglobulin; abs, antibodies; HCWs, health care workers; IFN, interferon; IGRA, IFN-γ release assay; and N, number.

**Figure 3 ijms-24-08525-f003:**
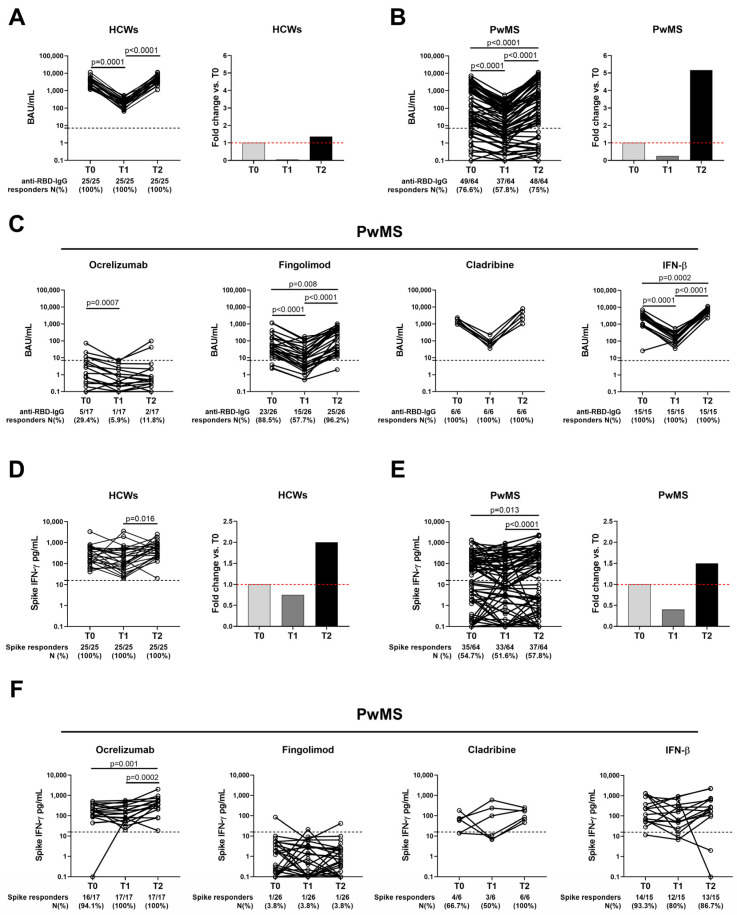
Kinetics of humoral- and IFN-γ-specific T-cell responses to COVID-19 vaccine in HCWs and PwMS. Antibody response and T-cell response detected by IGRA were assessed in 25 HCWs (**A**,**D**) and 64 PwMS (**B**,**E**) that were longitudinally sampled after 2–4 weeks (T0) and 6 months (T1) from the first vaccine dose and after 4–6 weeks from the booster dose (T2). Panels (**A**,**B**,**D**,**E**) reported on the left the histograms with the fold change calculated with respect to T0. (**C**,**F**) PwMS were stratified according to the ongoing therapy: ocrelizumab (*n* = 17), fingolimod (*n* = 26), cladribine (*n* = 6) and IFN-β (*n* = 15). Anti-RBD-IgG were detected in serum and reported as binding antibody units (BAU)/mL. IFN-γ levels were quantified in plasma harvested from stimulated samples by automatic ELISA and reported after subtracting the unstimulated-control value. Black dashed lines indicate the cut-off of the tests (anti-RBD-IgG: 7.1 BAU/mL and spike: 16 pg/mL). Red dashed lines indicate the starting response level. Friedman test was performed with Bonferroni correction (α/3) and a *p* < 0.016 was considered significant. Abbreviations: COVID-19, COronaVIrus Disease 2019; PwMS, patients with multiple sclerosis; RBD, receptor-binding domain; IgG, immunoglobulin; HCWs, health care workers; IFN, interferon; IGRA, IFN-γ release assay; N, number.

**Table 1 ijms-24-08525-t001:** Demographic and clinical characteristics of the 220 enrolled subjects.

Characteristics		PwMS	HCWs	*p* Value
**N (%)**		121 (55)	99 (45)	
**Age median (IQR)**		47 (39–55)	43 (31–52)	0.001 *
**Female N (%)**		85 (70.2)	72 (72.7)	0.685 ^§^
**Origin N (%)**	**West Europe**	118 (97.5)	97 (98)	0.531 ^§^
	**East Europe**	2 (1.7)	1 (1.0)	
	**Asia**	0 (0)	1 (1.0)	
	**South America**	1 (0.8)	0 (0)	
**BMI (kg/mq), median (IQR)**		23.7 (21.2–26.4)	-	
**MS duration, median (IQR)**		14 (8–22)	-	
**MS course N (%)**	**Relapsing-remitting**	110 (90)	-	
	**Primary-progressive**	11 (10)	-	
**EDSS score, median (IQR)**		2 (1–4)	-	
**Multiple sclerosis treatment N (%)**	**Ocrelizumab**	34 (28.1)	-	
	**Fingolimod**	35 (28.9)	-	
	**Cladribine**	20 (16.5)	-	
	**IFN-β**	32 (26.5)	-	
**Lymphocytes count N (%)**	**Available**	105 (86.8)	0 (0)	
**Lymphocytes count N (%)** **Median × 10^3^/µL (IQR)**	**Ocrelizumab**	34 (32.4) 1.62 (1.28–1.89)	-	<0.0001 **
	**Fingolimod**	35 (33.3) 0.65 (0.57–0.91)	-	
	**Cladribine**	19 (18.1) 1.10 (0.87–1.45)	-	
	**IFN-β**	17 (16.2) 1.65 (1.53–2.06)	-	
**Time between withdrawals, median days (IQR)**	**T0-T1**	161 (159–163)	139 (135–141)	
	**T1-T2**	63 (52–77)	168 (159–178)	
	**T0-T2**	218 (210–235)	305 (298–316)	

**Footnotes:** HCWs, health care workers; PwMS, patients with multiple sclerosis; IFN, interferon; BMI, body mass index; EDSS, Expanded Disability Status Scale; IQR, interquartile range; N, number. * Mann–Whitney U-statistic test; ^§^ Chi-square test; ** Kruskal–Wallis test performed only on MS patients.

**Table 2 ijms-24-08525-t002:** Serological and T-cell-specific responses at T1.

	Characteristics					*p* Value	
				PwMS	HCWs	Within MS Cohort	PwMS vs. HCWs
	**N (%)**			105 (54.1)	89 (45.9)		
**Antibody response**	**Qualitative response**	**anti-RBD abs responders N (%)**		69 (65.7)	89 (100)		<0.0001 ^§^
		**anti-RBD abs responders within the subgroups** **N (%)**	**Ocrelizumab**	6/29 (20.7)	-	<0.0001 ^§^	**<0.0001** ^§^
			**Fingolimod**	18/31 (58.1)	-		**<0.0001** ^§^
			**Cladribine**	18/18 (100)	-		-
			**IFN-β**	27/27 (100)	-		-
	**Quantitative response**	**anti-RBD titers** **BAU/mL median (IQR)**		32 (2.4–104)	177 (95.5–262.6)		<0.0001 *
			**Ocrelizumab**	0.80 (0.15–6.25)	-	<0.0001 ^#^	**<0.0001** *
			**Fingolimod**	7.9 (2.7–39.7)	-		**<0.0001** *
			**Cladribine**	85.9 (48–137)	-		**0.002** *
			**IFN-β**	126.4 (80–184)	-		0.047 *
**Spike-specific IFN-γ T-cell response**	**Qualitative response**	**spike responders N (%)**		63 (60)	88 (98.9)		<0.0001 ^§^
		**spike responders within the subgroups** **N (%)**	**Ocrelizumab**	28/29 (96.5)	-	<0.0001 ^§^	0.399 ^§^
			**Fingolimod**	1/31 (3.2)	-		**<0.0001** ^§^
			**Cladribine**	11/18 (61.1)	-		**<0.0001** ^§^
			**IFN-β**	23/27 (85.2)	-		**0.002** ^§^
	**Quantitative response**	**spike IFN-γ levels** **pg/mL median (IQR)**		39.9 (3.4–142.5)	218 (93–378)		<0.0001 *
			**Ocrelizumab**	114 (64–193)	-	<0.0001 ^#^	**0.006** *
			**Fingolimod**	0.8 (0.1–3.3)	-		**<0.0001** *
			**Cladribine**	32 (7–139)	-		**<0.0001** *
			**IFN-β**	79 (33–237)	-		**0.003** *

**Footnotes:** HCWs, health care workers; PwMS, patients with multiple sclerosis; N, number; IQR, interquartile range; ^§^ Chi-square test; * Mann–Whitney U-statistic test; ^#^ Kruskal–Wallis test; abs, antibodies; RBD, receptor-binding domain; S, spike. In bold are only those values that were significant after multiplicity correction by the Bonferroni method (α/4 = 0.0125).

**Table 3 ijms-24-08525-t003:** Final binomial regression models for demographic and clinical factors affecting the antibody (RBD) or T-cell (spike) immune responses after the second mRNA vaccine dose (T1) or after the booster dose (T2).

PwMS vs. HCWs						
RBD	T1			T2		
	**IRR**	**95%CI**	** *p* **	**IRR**	**95%CI**	** *p* **
HCWs	1.00			1.00		
IFN-β	1.96	0.78–4.91	0.149	2.92	1.32–6.44	**0.008**
Cladribine	1.19	0.48–2.92	0.708	1.67	0.59–4.77	0.334
Fingolimod	0.32	0.14–0.75	**0.009**	0.14	0.07–0.28	**<0.001**
Ocrelizumab	0.05	0.02–0.12	**<0.001**	0.09	0.05–0.18	**<0.001**
Weeks elapsed from the vaccination	0.82	0.68–0.97	**<0.001**	0.72	0.61–0.86	**<0.001**
**Within PwMS cohort**						
**RBD**		**T1**			**T2**	
	**IRR**	**95%CI**	** *p* **	**IRR**	**95%CI**	** *p* **
IFN-β	1.00			1.00		
Cladribine	0.78	0.29–2.11	0.624	0.50	0.08–3.20	0.465
Fingolimod	0.23	0.10–0.53	**0.001**	0.04	0.01–0.16	**<0.001**
Ocrelizumab	0.04	0.02–0.12	**<0.001**	0.03	0.01–0.22	**<0.001**
Treatment duration	1.02	0.95–1.09	0.585	0.98	0.88–1.10	0.776
EDSS	0.96	0.84–1.09	0.488	1.08	0.84–1.39	0.546
**PwMS vs. HCWs**						
**Spike**	**T1**			**T2**		
	**IRR**	**95%CI**	** *p* **	**IRR**	**95%CI**	** *p* **
HCWs	1.00			1.00		
IFN-β	0.78	0.17–3.45	0.743	0.55	0.27–1.12	0.098
Cladribine	0.33	0.07–1.54	0.163	0.12	0.05–0.30	**<0.001**
Fingolimod	0.01	0.002–0.04	**<0.001**	0.003	0.002–0.01	**<0.001**
Ocrelizumab	0.73	0.18–2.96	0.662	0.48	0.24–0.96	**0.037**
Weeks elapsed from the vaccination	0.91	0.67–1.25	0.586			
Age	0.97	0.96–0.99	**0.009**			
**Within PwMS cohort**						
**Spike**	**T1**			**T2**		
	**IRR**	**95%CI**	** *p* **	**IRR**	**95%CI**	** *p* **
IFN-β	1.00			1.00		
Cladribine	0.68	0.26–1.75	0.421	0.23	0.08–0.72	**0.012**
Fingolimod	0.02	0.01–0.04	**<0.001**	0.01	0.001–0.01	**<0.001**
Ocrelizumab	1.13	0.49–2.57	0.778	0.90	0.38–2.15	0.814
Lymphocytes	1.01	0.74–1.38	0.939	0.92	0.61–1.40	0.709

**Footnotes:** IRR: incidence rate ratio; PwMS: patients with multiple sclerosis; HCWs: health care workers; CI: confidence interval; RBD: receptor-binding domain; IFN, interferon; EDSS: Expanded Disability Status Scale. In bold are reported the significant values.

**Table 4 ijms-24-08525-t004:** Serological and T-cell-specific responses at T2.

	Characteristics					*p* Value	
				PwMS	HCWs	Within MS Cohort	PwMS vs. HCWs
	**N (%)**			82 (68.3)	38 (31.7)		
**Antibody response**	**Qualitative response**	**anti-RBD abs responders** **N (%)**		64 (78)	38 (100)		0.0017 ^§^
		**anti-RBD abs responders within the subgroups** **N (%)**	**Ocrelizumab**	5/22 (22.7)	-	<0.0001 ^§^	**<0.0001** ^§^
			**Fingolimod**	29/30 (96.7)	-		0.257 ^§^
			**Cladribine**	10/10 (100)	-		-
			**IFN-β**	20/20 (100)	-		-
	**Quantitative response**	**anti-RBD titers** **BAU/mL median (IQR)**		408 (11.6–4235)	4516 (3098–5477)		<0.0001 *
			**Ocrelizumab**	0.55 (0.27–6.27)	-	<0.0001 ^#^	**<0.0001 ***
			**Fingolimod**	174 (26–514)	-		**<0.0001 ***
			**Cladribine**	3455 (1300–6060)	-		0.295 *
			**IFN-β**	6060 (3906–9414)	-		0.023 *
**Spike-specific IFN-γ T-cell response**	**Qualitative response**	**spike responders N (%)**		49 (59.7)	38 (100)		<0.0001 ^§^
		**spike responders within the subgroups** **N (%)**	**Ocrelizumab**	22/22 (100)	-	<0.0001 ^§^	-
			**Fingolimod**	1/30 (3.3)	-		**<0.0001** ^§^
			**Cladribine**	8/10 (80)	-		**0.005** ^§^
			**IFN-β**	18/20 (90)	-		0.047 ^§^
	**Quantitative response**	**spike IFN-γ levels** **pg/mL median (IQR)**		71.9 (1.2–334.5)	426 (196–792)		<0.0001 *
			**Ocrelizumab**	352 (230–544)	-	<0.0001 ^#^	0.364 *
			**Fingolimod**	0.6 (0.1–2.6)	-		**<0.0001 ***
			**Cladribine**	75 (35–176)	-		**<0.0001 ***
			**IFN-β**	215 (99–728)	-		0.084 *

**Footnotes:** HCWs, health care workers; PwMS, patients with multiple sclerosis; N, number; IQR, interquartile range; ^§^ Chi-square test; * Mann–Whitney U-statistic test; ^#^ Kruskal–Wallis test; abs, antibodies; RBD, receptor-binding domain; IFN, interferon. In bold are only those values that were significant after multiplicity correction by the Bonferroni method (α/4 = 0.0125).

**Table 5 ijms-24-08525-t005:** Longitudinal observation of T-cell and antibody responses.

			T0	T1	T2	*p* Value
**HCWs**		**N (%)**	25 (100%)	25 (100%)	25 (100%)	
	**Qualitative response**	**Antibody responders N (%)**	25/25 (100%)	25/25 (100%)	25/25 (100%)	-
		**T-cell responders N (%)**	25/25 (100%)	25/25 (100%)	25/25 (100%)	-
	**Quantitative response**	**anti-RBD titers** **BAU/mL median (IQR)**	3377 (1647–4839)	178 (132–271)	4608 (3040–6408)	<0.0001 ^#^
		**spike IFN-γ levels** **pg/mL median (IQR)**	262 (118–531)	196 (66–490)	525 (234–812)	0.012 ^#^
**PwMS**		**N (%)**	64 (100%)	64 (100%)	64 (100%)	
	**Qualitative response**	**Antibody responders** **N (%)**	49/64 (76.6%)	37/64 (57.8%)	48/64 (75%)	<0.0001 *
		**T-cell responders** **N (%)**	35/64 (54.7%)	33/64 (51.6%)	37/64 (57.8%)	0.368 *
	**Quantitative response**	**anti-RBD titers** **BAU/mL median (IQR)**	52.8 (7.8–1165)	13.2 (1.6–98.6)	272.5 (6.0–4558)	<0.0001 ^#^
		**spike IFN-γ levels** **pg/mL median (IQR)**	52.4 (1.7–178)	20.8 (1.7–171)	78.6 (1.0–351)	0.031 ^#^

**Footnotes:** N, number; IQR, interquartile range; RBD, receptor-binding domain; HCWs, health care workers; PwMS, patients with multiple sclerosis. * Cochran’s Q test and ^#^ Friedman test were performed for the statistical analysis.

## Data Availability

The raw data generated and/or analyzed in the present study are available in our institutional repository (rawdata.inmi.it), subject to registration. The data can be found by selecting the article of interest from a list of articles ordered by the year of publication. No charge for granting access to data is required. In the event of a malfunction of the application, the request can be sent directly by e-mail to the Library (biblioteca@inmi.it).

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
