# Peer review of "Dynamic Evolution of Humoral and T-Cell Specific Immune Response to COVID-19 mRNA Vaccine in Patients with Multiple Sclerosis Followed until the Booster Dose"

_ijms, 2023, doi:10.3390/ijms24108525_

Round 1

Reviewer 1 Report

The article entitled as "Dynamic evolution of humoral and T-cell specific immune response to COVID-19 mRNA vaccine in patients with multiple sclerosis followed until the booster dose" is an interesting piece of work as it is talking about the changing patterns of the immune response. Particularly this work is more of importance amid the evolution of SARS-CoV-2. This work can be potential guidelines to decide the fate of booster doses especially among the susceptible population.

I congratulate the authors for such important work. 

However, I suggest considering the following points before the publication.

I found introduction non satisfactory and suggest improving the introduction with recent references.  

Line no. 56:  COronaVirus 2, change Coronavirus.

https://doi.org/10.3390/vaccines11010101

I wonder where is the Figure 3?

Figure 3. Kinetics of humoral and IFN-γ-specific T-cell responses to COVID-19 vaccine in HCWs 

The discussion needs to be improved to justify that the results you obtained are scientifically sound. See if you can find any related studies published in 2023. The same can be incorporated. 

Personally, as a reader I would like to read the conclusions separately as a section. Which will make easy to follow all the major findings of the research. If possible, make changes accordingly. 

Best Wishes

The English used is acceptable and needs final check before the publication. 

Author Response

Reviewer #1:

The article entitled as "Dynamic evolution of humoral and T-cell specific immune response to COVID-19 mRNA vaccine in patients with multiple sclerosis followed until the booster dose" is an interesting piece of work as it is talking about the changing patterns of the immune response. Particularly this work is more of importance amid the evolution of SARS-CoV-2. This work can be potential guidelines to decide the fate of booster doses especially among the susceptible population.

I congratulate the authors for such important work. 

AUTHORS: We sincerely thank the reviewer for the comment.

However, I suggest considering the following points before the publication.

I found introduction non satisfactory and suggest improving the introduction with recent references.  

AUTHORS: We thank the reviewer for the observation. We updated the introduction with the following recent references: Sunagar et al., Vaccines, 2023; Dhawan et al., Vaccines, 2023; Sansone et al., Vaccines, 2023; Gillot et al., Clin Chem Lab Med 2023; Santoro et al., Biomedicines, 2023; Menegale et al., JAMA Netw Open, 2023; Sainz de la Maza et al., Vaccines, 2023; Achiron et al., Mult Scler Relat Disord, 2023; Katz Sand et al., Mult Scler Relat Disord, 2023; Torres et al., J Neurol, 2023; Lambrianides et al., Front Neurol, 2023. We also added information regarding the different vaccines options as required by the reviewer 2.

Line no. 56:  COronaVirus 2, change Coronavirus.

AUTHORS: We thank the reviewer for the observation. We modified it accordingly.

I wonder where is the Figure 3?

Figure 3. Kinetics of humoral and IFN-γ-specific T-cell responses to COVID-19 vaccine in HCWs 

AUTHORS: The figure is at page 10.

The discussion needs to be improved to justify that the results you obtained are scientifically sound. See if you can find any related studies published in 2023. The same can be incorporated. 

AUTHORS: We thank the reviewer for raising this point. We included in the discussion recent references (Sainz de la Maza et al., Vaccines, 2023; Achiron et al., Mult Scler Relat Disord, 2023; Conway et al., Mult Scler J Exp Transl Clin, 2023) to further support our findings.

Personally, as a reader I would like to read the conclusions separately as a section. Which will make easy to follow all the major findings of the research. If possible, make changes accordingly. 

AUTHORS: We thank the reviewer for the suggestion. We separate sections for limitations and conclusions as also required by the reviewer 2.

We sincerely thank the Reviewer for the time dedicated to our manuscript and we hope that this manuscript is now suitable for publication in “International Journal of Molecular Sciences”.

Best regards

Delia Goletti, MD, PhD

Clinical Investigator

Head of Translational Research Unit of the Research Department

Padiglione del Vecchio, Room 39

National Institute for Infectious Diseases

Via Portuense 292, Rome 00149, Italy

Tel: +39-06-55170-906; Fax: +39-06-5582-825

E-mail address: [email protected];

Reviewer 2 Report

1.       Why authors specifically chose the mRNA vaccine? What about other vaccines? Please justify and discuss.

2.       What are the future impact of this work? This should be stated in the abstract.

3.       There are no males in this study. Please justify and discuss.

4.       Make separate sections for limitations and conclusions.

5.       Did the authors make sure that subjects are not under any other medication during the experiment time? Please justify and discuss.

6.       During COVID-19 people are using many dietary supplements to improve immunity. Did the authors collect all relevant information?

7.       The authors should also discuss the other types of vaccines and their reported impact on immunity.

There are some grammatical mistakes and wrongly used phrases. Please checkout and rewrite wherever required. 

Author Response

To Ms. Vicky Shi,

Assistant Editor of the International Journal of Molecular Sciences 

Rome, May 5th, 2023 

Dear Editor,  

Thank you for the opportunity to send you the revised version of the manuscript submitted to the “International Journal of Molecular Sciences”, Manuscript ID: ijms-2384207, by Ruggieri and Aiello et al. 

Below, please find the point-by-point answers to the reviewer 2. 

Reviewer #2:

Comments and Suggestions for Authors

  1. Why authors specifically chose the mRNA vaccine? What about other vaccines? Please justify and discuss.

AUTHORS: We thank the reviewer for raising this point. This study was conducted on HCWs and PwMS who received mRNA-vaccines because in Italy most of the population was vaccinated with BNT162b2 or mRNA-1273. Moreover, since the enrolment of health care workers and PwMS began in January and March 2021 respectively, the mRNA-based vaccines were the first available at that time. The discussion about other vaccines has been included in the introduction.

  1. What are the future impact of this work? This should be stated in the abstract.

AUTHORS: We thank the reviewer for raising this point. We stated the impact of this work in the abstract.

  1. There are no males in this study. Please justify and discuss.

AUTHORS: We thank the reviewer for the comment. As shown in Table 1, the percentage of female is 70.2% in PwMS and 72% in HCWs. This is due to the fact the multiple sclerosis is a disease affecting women 3 times more frequently than men (Harbo et al., Ther Adv Neurol Disord, 2013, doi:10.1177/1756285613488434). Regarding HCWs, in Italy approximately 68% of the staff in the National Health Service is represented by women. However, the two cohorts are matched for gender.

This comment has been added in the text (lines 129-131).

  1. Make separate sections for limitations and conclusions.

AUTHORS: We thank the reviewer for the suggestion. We modified it accordingly.

  1. Did the authors make sure that subjects are not under any other medication during the experiment time? Please justify and discuss.

AUTHORS: We thank the reviewer for this comment. Patients didn’t undergo any other therapy that could influence their immunological state through the study period. This was specified in the method section (pag. 14 line 431-432) as “ongoing DMT with ocrelizumab, fingolimod, cladribine or IFN-β for at least 6 months prior to the enrollment”. To be more consistent, we checked all the concomitant medications at each patients’ encounter.

  1. During COVID-19 people are using many dietary supplements to improve immunity. Did the authors collect all relevant information?

AUTHORS: We thank the reviewer for this observation. Even though we did not check specifically this information, we can exclude our patients with MS were taking dietary supplement. Patients with MS do not take any dietary supplements without treating neurologist approval. Moreover, we checked the concomitant medications (please see previous comment).

7.The authors should also discuss the other types of vaccines and their reported impact on immunity.

AUTHORS: We thank the reviewer for the comment. We included it in the introduction and in the discussion section.

Comments on the Quality of English Language

There are some grammatical mistakes and wrongly used phrases. Please check-out and rewrite wherever required. 

AUTHORS: We thank the reviewer for the suggestion. We checked-out the text.

We sincerely thank the Reviewer for the time dedicated to our manuscript and we hope that this manuscript is now suitable for publication.

Best regards

Delia Goletti, MD, PhD

Clinical Investigator

Head of Translational Research Unit of the Research Department

Padiglione del Vecchio, Room 39

National Institute for Infectious Diseases

Via Portuense 292, Rome 00149, Italy

Tel: +39-06-55170-906; Fax: +39-06-5582-825

E-mail address: [email protected];

Round 2

Reviewer 1 Report

I appreciate the efforts of the authors to revise the manuscript. The manuscript is sufficiently revised and can be accepted in the present form. 

Best Wishes  

The English used is acceptable, however, I recommend to double check the manuscript for any minor grammatical errors. 

Reviewer 2 Report

The authors successfully responded to the reviewer's comments and updated the manuscript as well.

It is acceptable.